

# Downscaling surface wind predictions from numerical weather prediction models in complex terrain with WindNinja

N. S. Wagenbrenner[1], J. M. Forthofer[1], B. K. Lamb[2], K. S. Shannon[1], and B. W. Butler[1]

[1]US Forest Service, Rocky Mountain Research Station, Missoula Fire Sciences Laboratory, 5775 W Highway 10, Missoula, MT 59808, USA
[2]Laboratory for Atmospheric Research, Department of Civil and Environmental Engineering, Washington State University, Pullman, WA 99164, USA

Received: 25 September 2015 – Accepted: 6 December 2015 – Published: 18 January 2016

Correspondence to: N. S. Wagenbrenner (nwagenbrenner@fs.fed.us)

Published by Copernicus Publications on behalf of the European Geosciences Union.



## Abstract

Wind predictions in complex terrain are important for a number of applications. Dynamic downscaling of numerical weather prediction (NWP) model winds with a high resolution wind model is one way to obtain a wind forecast that accounts for local terrain effects, such as wind speed-up over ridges, flow channeling in valleys, flow separation around terrain obstacles, and flows induced by local surface heating and cooling. In this paper we investigate the ability of a mass-consistent wind model for downscaling near-surface wind predictions from four NWP models in complex terrain. Model predictions are compared with surface observations from a tall, isolated mountain. Downscaling improved near-surface wind forecasts under high-wind (near-neutral atmospheric stability) conditions. Results were mixed during upslope and downslope (non-neutral atmospheric stability) flow periods, although wind direction predictions generally improved with downscaling. This work constitutes evaluation of a diagnostic wind model at unprecedented high spatial resolution in terrain with topographical ruggedness approaching that of typical landscapes in the western US susceptible to wildland fire.

## 1 Introduction

Researchers from multiple disciplines rely on routine forecasts from numerical weather prediction (NWP) models to drive transport and dispersion models, conduct wind assessments for wind energy projects, and predict the spread of wildfires. These applications require fine-scale, near-surface wind predictions in regions where rugged terrain and vegetation have a significant effect on the local flow field. Terrain effects such as wind speed-up over ridges, flow channeling in valleys, flow separation around terrain obstacles, and enhanced surface roughness alter the flow field over spatial scales finer than those used for routine, operational NWP forecasting.

Numerous operational mesoscale NWP model forecast products are available in real-time, such as those provided by National Centers for Environmental Prediction

Discussion Paper | Discussion Paper | Discussion Paper | Discussion Paper |

# ACPD

doi:10.5194/acp-2015-761

**Downscaling surface wind predictions with WindNinja**

N. S. Wagenbrenner et al.

(NCEP). Access to these output products is facilitated by automated archiving and distribution systems such as the National Operational Model Archive and Distribution System (NOMADS). These routine forecast products are highly valuable to researchers and forecasters, for example, as inputs to drive other models. In many cases, however,
the spatial resolution of the system of interest (e.g., wildland fire spread) is much finer than that of the NWP model output.

The model grid horizontal resolution in operational NWP models is limited due, in part, to the high computational demands of NWP. Routine gridded forecast products are typically provided at grid resolutions of 3 km or larger. The High Resolution Rapid Re-
10 fresh (HRRR) model produces 3 km output grids and is currently the highest-resolution operational forecast in the US.

NWP models have been run successfully with grid resolutions of less than 1 km in complex terrain for specific cases when modifications were made to the meshing (Lundquist et al., 2010) or PBL schemes (Ching et al., 2014; Seaman et al., 2012) or
15 when large-eddy simulation (LES) was used (Chow and Street, 2008). While successful for specific test cases, these efforts employ specialized model configurations that have not been incorporated into routine forecasting frameworks, either because they are not sufficiently robust, have not been thoroughly tested, or are too computationally intense for routine forecasting. For example, the configuration used in Seaman et al. (2012) is
20 applicable for stable nocturnal conditions only.

Additionally, these modifications require technical expertise in NWP and access to substantial computing resources, which many consumers of NWP output do not have. Perhaps, the biggest limitation to running NWP models on grids with fine horizontal resolution is the computational demand. Time-sensitive applications, such as operational
wildland fire support, require fast solution times (e.g., less than 1 h) on simple hardware (e.g., laptop computers with 1–2 processors). Thus, there remains a practical need for fast-running tools that can be used to downscale coarse NWP model winds in complex terrain.

**ACPD**

doi:10.5194/acp-2015-761

**Downscaling surface wind predictions with WindNinja**

N. S. Wagenbrenner et al.

Dynamic downscaling with a steady-state (diagnostic) wind model is one option for obtaining near-surface high-resolution winds from routine NWP model output (e.g., Beaucage et al., 2014). The NWP model provides an initial wind field that accounts for mesoscale dynamics which is then downscaled by a higher resolution wind model
to enforce conservation of mass and, in some cases, momentum and energy on the flow field on a higher resolution grid that better resolves individual terrain features. Dynamic downscaling can be done in a steady-state fashion for each time step of the NWP model output. One advantage of using a steady-state downscaling approach is that the spatial resolution can be increased with no additional computational cost associated
with an increase in temporal resolution.

Diagnostic wind models have primarily been evaluated with observations collected over relatively simple, low elevation hills. Askervein Hill (Taylor and Teunissen, 1987) and Bolund Hill (Berg et al., 2011) are the two mostly commonly used datasets for evaluating diagnostic wind models. These are both geometrically simple, low-elevation
hills compared to the complex terrain exhibited in many regions of the western US susceptible to wildland fire. Lack of evaluations under more complex terrain is due in part to the lack of high-resolution datasets available in complex terrain. Recently, Butler et al. (2015) reported high-resolution wind observations from a tall, isolated mountain (Big Southern Butte) in the western US. Big Southern Butte is substantially taller and
more geometrically complex than both Askervein and Bolund hills.

In this work, we investigate the ability of a mass-conserving wind model, WindNinja (Forthofer et al., 2014a), for dynamically downscaling NWP model winds over Big Southern Butte. WindNinja is a diagnostic wind model developed for operational wildland fire support. It has previously been evaluated against the Askervein Hill data
(Forthofer et al., 2014a) and found to capture important terrain-induced flow features, such as ridgetop speed-up, and it has been shown to improve wildfire spread predictions in complex terrain (Forthofer et al., 2014b).

The goals of this work were to (1) investigate the accuracy of NWP model near-surface wind predictions in complex terrain on spatial scales relevant for processes

**ACPD**

doi:10.5194/acp-2015-761

**Downscaling surface wind predictions with WindNinja**

N. S. Wagenbrenner et al.

driven by local surface winds, such as wildland fire behavior and (2) assess the ability of a mass-consistent wind model to improve these predictions through dynamic down-scaling. Wind predictions are investigated from four NWP models operated on different horizontal grid resolutions. This work constitutes one of the first evaluations of a di-
agnostic wind model with data collected over terrain with a topographical ruggedness approaching that of western US landscapes susceptible to wildland fire.

## 2   Model descriptions and configurations

WRF is a NWP model that solves the non-hydrostatic, fully compressible Navier–Stokes equations using FDM discretization techniques (Skamarock et al., 2008). All
of the NWP models investigated in this work use either the Advanced Research WRF (ARW) or the non-hydrostatic multi-scale model (NMM) core of the WRF model (Table 1).

### 2.1   Routine Weather Research and Forecasting (WRF-UW)

Routine WRF-ARW forecasts with 4 km horizontal grid resolution were acquired from
the University of Washington Atmospheric Sciences forecast system (www.atmos. washington.edu/mm5rt/info.html). These forecasts are referred to as WRF-UW. The outer domain of WRF-UW has a horizontal grid resolution of 36 km and covers most of the western US and northeastern Pacific Ocean. This outer domain is initialized with NCEP Global Forecast System (GFS) 1° runs. The 36 km grid is nested down
to 12, 4 km, and an experimental 1.33 km grid which covers a limited portion of the Pacific Northwest. The 4 km grid investigated in this study covers the Pacific North-west, including Washington, Oregon, Idaho, and portions of California, Nevada, Utah, Wyoming, and Montana. Physical parameterizations employed by WRF-UW include the Noah Land Surface Model (Chen et al., 1996), Thompson microphysics (Thomp-
son et al., 2004), Kain (2004) for convection, Rapid Radiative Transfer Model (RRTM)

**ACPD**

doi:10.5194/acp-2015-761

**Downscaling surface wind predictions with WindNinja**

N. S. Wagenbrenner et al.

for longwave radiation (Mlawer et al., 1997), Duhdia (1989) for shortwave radiation, and the Yonsei University (YSU) boundary layer scheme (Hong et al., 2006). WRF-UW is run at 00z and 12z and generates hourly forecasts out to 84 h. The computational domain consists of 38 vertical layers. The first grid layer is approximately 40 m a.g.l. and the average model top height is approximately 16 000 m a.g.l.

## 2.2 Weather Research and Forecasting Reanalysis (WRF-NARR)

WRF-ARW reanalysis runs were performed using the NCEP North American Regional Reanalysis (NARR) data (Mesinger et al., 2006). The reanalysis runs are referred to as WRF-NARR. The same parameterizations and grid nesting structures used in WRF-UW were also used for the WRF-NARR simulations, except that the WRF-NARR inner domain had 33 vertical layers and a horizontal grid resolution of 1.33 km (Table 1). Analysis nudging (e.g., Stauffer and Seaman, 1994) was used above the boundary layer in the outer domain (36 km horizontal grid resolution). Hourly WRF-NARR simulations were run for 15 day periods with 12 h of model spin up prior to each simulation. The first grid layer was approximately 38 m a.g.l. and the average model top height was approximately 15 000 m a.g.l. WRF-NARR differs from the other models used in this study in that it is not a routinely run model. These were custom simulations conducted by our group to provide a best-case scenario for the NWP models. Routine forecasts are already available for limited domains (e.g., UW provides WRF simulations on a 1.33 km grid for a small domain in the Pacific Northwest of the US) and are likely to become more widely available at this grid resolution in the near future.

## 2.3 North American Mesoscale Model (NAM)

The North American Mesoscale (NAM) model is an operational forecast model run by NCEP for North America (http://www.emc.ncep.noaa.gov/index.php?branch=NAM). The NAM model uses the NMM core of the WRF model. The NAM CONUS domain investigated in this study has a horizontal grid resolution of 12 km. NAM employs the

**ACPD**

doi:10.5194/acp-2015-761

**Downscaling surface wind predictions with WindNinja**

N. S. Wagenbrenner et al.

Noah Land Surface model (Chen et al., 1996), Ferrier et al. (2003) for microphysics, Kain (2004) for convection, GFDL (Lacis and Hansen, 1974) for longwave and short-wave radiation, and the Mellor–Yamada–Janjic (MJF) boundary layer scheme (Janjic, 2002). The NAM model is initialized with 12 h runs of the NAM Data Assimilation System. It is run four times daily at 00z, 06z, 12z, and 18z and generates hourly forecasts out to 84 h. The computational domain consists of 26 vertical layers. The first grid layer is approximately 200 m a.g.l. and the average model top height is approximately 15 000 m a.g.l. NAM forecasts are publicly available in real time from NCEP.

## 2.4 High Resolution Rapid Refresh (HRRR)

The High Resolution Rapid Refresh (HRRR) system is a nest inside of the NCEP-Rapid Refresh (RAP) model (13 km horizontal grid resolution; http://ruc.noaa.gov/hrrr/). HRRR has a horizontal grid resolution of 3 km and is updated hourly. HRRR uses the WRF model with the ARW core and employs the RUC-Smirnova Land Surface Model (Smirnova et al., 1997, 2000), Thompson et al. (2004) microphysics, RRTM longwave radiation (Mlawer et al., 1997), Goddard shortwave radiation (Chou and Suarez, 1994), the MYJ boundary layer scheme (Janjic, 2002). HRRR is initialized from 3 km grids with 3 km radar assimilation over a 1 h period. HRRR is currently the highest resolution operational forecast available in real time. The computational domain consists of 51 vertical layers. The first grid layer is approximately 8 m a.g.l. and the average model top height is approximately 16 000 m a.g.l.

## 2.5 WindNinja

WindNinja is a mass-conserving diagnostic wind model developed and maintained by the USFS Missoula Fire Sciences Laboratory (Forthofer et al., 2014a). The theoretical formulation is described in detail in Forthofer et al. (2014a). Here we provide a brief overview of the modeling framework. WindNinja uses a variational calculus technique to minimize the change in an initial wind field while conserving mass locally (within each

**ACPD**

doi:10.5194/acp-2015-761

**Downscaling surface wind predictions with WindNinja**

N. S. Wagenbrenner et al.

cell) and globally over the computational domain. The numerical solution is obtained using finite element method (FEM) techniques on a terrain-following mesh consisting of layers of hexahedral cells that grow vertically with height.

WindNinja includes a diurnal slope flow parameterization (Forthofer et al., 2009). The diurnal slope flow model used in WindNinja is the shooting flow model in Mahrt (1982). It is a one-dimensional model of buoyancy-driven flow along a slope. A micrometeorological model similar to the one used in CALMET (Scire et al., 2000; Scire and Robe, 1997) is used to compute surface heat flux, Monin–Obukhov length, and boundary layer height. The slope flow is then calculated as a function of sensible heat flux, distance to ridgetop or valley bottom, slope steepness, and surface and entrainment drag parameters. The slope flow is computed for each grid cell and added to the initial wind in that grid cell. Additional details can be found in Forthofer et al. (2009).

WindNinja was used to dynamically downscale hourly 10 m wind predictions from the above NWP models. The WindNinja computational domain was constructed from 30 m resolution Shuttle Radar Topography Mission (SRTM) data (Farr et al., 2007). The 10 m NWP winds were bilinearly interpolated to the WindNinja computational domain and used as the initial wind field. Layers above and below the 10 m height were fit to a logarithmic profile based on the micrometeorological model. The computational domain consisted of 20 vertical layers. The first grid layer is 1.92 m a.g.l. and the average model top height is 931 m a.g.l.

## 2.6 Terrain representation

The four NWP models used in this study employ an implementation of the WRF model. They use different initial and boundary conditions, incorporate different parameterizations for sub-grid processes, such as land surface fluxes, convection, and PBL evolution, but in terms of surface wind predictions under the conditions investigated in this study (inland, dry summertime conditions), the horizontal grid resolution is arguably the most important difference among the models. The horizontal grid resolution affects the numerical solution since fewer terrain features are resolved by coarser grids. Coarser

grids essentially impart a smoothing effect which distorts the actual geometry of the underlying terrain (Fig. 1). As horizontal cell size and terrain complexity increase, the accuracy of the terrain representation and thus, the accuracy of the near-surface flow solution deteriorate.

## 3   Evaluations with field observations

### 3.1   Observations at Big Southern Butte

Surface wind data (Butler et al., 2015) collected from an isolated mountain (Big Southern Butte, hereafter "BSB"; 43.395958, −113.02257) in southeast Idaho were used to evaluate surface wind predictions (Fig. 1). BSB is a predominantly grass-covered volcanic cinder cone with a horizontal scale of 5 km and a vertical scale of 800 m and surrounded in all directions by the relatively flat Snake River Plain. The portion of the Snake River Plain surrounding BSB slopes downward gently from the northeast to the southwest.

Three-meter wind speeds and directions were measured with cup-and-vane anemometers at 53 locations on and around BSB. The anemometers have a measurement range of 0 to $44\,\mathrm{m\,s^{-1}}$, a resolution of $0.19\,\mathrm{m\,s^{-1}}$ and 1.4°, and are accurate to within $\pm 0.5\,\mathrm{m\,s^{-1}}$ and ±5°. The anemometers measured wind speed and direction every second and logged 30 s averages. We averaged these 30 s winds over a 10 min period at the top of each hour (five minutes before and 5 min after the hour). The 10 min averaging period was chosen to correspond roughly with the time scale of wind predictions from the NWP forecasts. The NWP output is valid at a particular instant in time, but there is always some inherent temporal averaging in the predictions. The temporal averaging associated with a given prediction depends on the time-step used in the NWP model and is typically on the order of minutes. The 10 min averaged observed data are referred to in the text as "hourly" observations (since they are averaged at the top of each hour) and are compared directly with the hourly model predictions.

**ACPD**

doi:10.5194/acp-2015-761

**Downscaling surface wind predictions with WindNinja**

N. S. Wagenbrenner et al.

Butler et al. (2015) observed the following general flow features at BSB. During periods of weak synoptic and mesoscale forcing (hereafter, referred to collectively as "external forcing"), the observed surface winds at BSB were decoupled from the large-scale atmospheric flows, except for at high-elevation ridgetop locations. Diurnal slope flows dominated the local surface winds under periods of weak external forcing. There were frequent periods of strong external forcing, during which the diurnal slope winds on BSB were completely overtaken by the larger-scale winds. These periods of strong external forcing at BSB were typically characterized by large-scale southwesterly flow aligned with the Snake River Plain, although occasionally there were also strong early morning winds from the northeast. Under periods of strong external forcing wind speeds commonly varied by as much as $15\,\mathrm{m\,s^{-1}}$ across the domain due to mechanical effects of the terrain (e.g., speed-up over ridges and lower speeds on leeward slopes). Additional details regarding the BSB field campaign can be found in Butler et al. (2015).

## 3.2 Evaluation methods

Hourly observations were compared against corresponding hourly predictions from the most recent model run. Modeled and observed winds were compared by interpolating the modeled surface wind variables to the observed surface sensor locations at each site. The 10 m winds from the NWP forecasts were interpolated to sensor locations, using bilinear interpolation in the horizontal dimension and a log profile in the vertical dimension. A 3-D interpolation scheme was used to interpolate WindNinja winds to the sensor locations. This 3-D interpolation was possible because the WindNinja domain had layers above and below the surface sensor height (3.0 m a.g.l.). A 3-D interpolation scheme was not possible for the NWP domains since there were not any layers below the three meter surface sensor height.

**ACPD**

doi:10.5194/acp-2015-761

**Downscaling surface wind predictions with WindNinja**

N. S. Wagenbrenner et al.

Model performance was quantified in terms of the mean bias, root-mean-square error (RMSE), and standard deviation of the error (SDE):

$$\overline{\varphi'} = \frac{1}{N} \sum_{i=1}^{N} \varphi' \tag{1}$$

$$\text{RMSE} = \left[ \frac{1}{N} \sum_{i=1}^{N} \left( \varphi'_i \right)^2 \right]^{1/2} \tag{2}$$

$$\text{SDE} = \left[ \frac{1}{N-1} \sum_{i=1}^{N} \left( \varphi'_i - \overline{\varphi'} \right)^2 \right]^{1/2} \tag{3}$$

where $\varphi'$ is the difference between simulated and observed variables and $N$ is the number of observations.

## 3.3 Case selection

We selected a five-day period from 15–19 July 2010 for model evaluations. This specific period was chosen because it included periods of both strong and weak external forcing, conditions were consistently dry and sunny, and was a period for which we were able to acquire forecasts from all NWP models selected for investigation in this study.

The observed data from the five-day period were broken into periods of upslope, downslope, and externally-driven flow conditions to further investigate model performance under these particular types of flow regimes. We used the partitioning schemes described in Butler et al. (2015). Externally-driven events were partitioned out by screening for hours during which wind speeds at a designated sensor (R2, located 5 km southwest of the butte in flat terrain) exceeded a predetermined threshold wind speed of $6\,\text{m s}^{-1}$. This sensor was chosen because it was located in flat terrain far from

Discussion Paper | Discussion Paper | Discussion Paper | Discussion Paper |

**ACPD**

doi:10.5194/acp-2015-761

**Downscaling surface wind predictions with WindNinja**

N. S. Wagenbrenner et al.

the butte and therefore was representative of near-surface winds that were largely un-affected by the butte itself. Hours of upslope and downslope flows (i.e., observations under weak external forcing) were then partitioned out of the remaining data. Additional details regarding the partitioning scheme can be found in Butler et al. (2015). Statistical metrics were computed for these five-day periods.

We also chose one specific hour representative of each flow regime within the 5 day period to qualitatively investigate model performance for single flow events under the three flow regimes. This direct comparison of NWP model predictions, downscaled predictions, and observations for single events was performed in order to get a visual sense of how the models performed spatially, while avoiding any inadvertent compli-cating issues that may have arisen from temporal averaging over the flow regimes.

## 4 Results and discussion

### 4.1 Overview of the five-day simulations

Figure 2 shows observed vs. forecasted wind speeds during the five-day period. The following generalizations can be made. The NWP models predicted wind speeds below $5\,\mathrm{m\,s^{-1}}$ reasonably well on average, although HRRR tended to over predict at speeds below $3\,\mathrm{m\,s^{-1}}$ (Fig. 2). There is a lot of scatter about the regression lines, but the re-gressions follow the line of agreement fairly well up to observed speeds around $5\,\mathrm{m\,s^{-1}}$. Downscaling did not improve wind speed predictions much in this range. NWP forecast accuracy declined for observed speeds between 5 and $10\,\mathrm{m\,s^{-1}}$, and accuracy sharply dropped off for observed speeds above $10\,\mathrm{m\,s^{-1}}$. This is indicated by the rapid depar-ture of the NWP model regression lines from the line of agreement (Fig. 2). Downscal-ing improved wind speed predictions for all NWP forecasts for observed speeds greater than around $5\,\mathrm{m\,s^{-1}}$ and the biggest improvements were for observed speeds greater than $10\,\mathrm{m\,s^{-1}}$ (Fig. 2). This is indicated by the relative proximity of the downscaled regression lines to the line of agreement (Fig. 2).

**ACPD**

doi:10.5194/acp-2015-761

**Downscaling surface wind predictions with WindNinja**

N. S. Wagenbrenner et al.

Poor model accuracy at higher speeds is largely due to the models under predicting windward slope and ridgetop wind speeds. Observed speeds at these locations were often three or four times higher than speeds in other locations in the study area (e.g., note the spatial variability in Fig. 3). Butler et al. (2015) showed that the highest observed speeds occurred on upper elevation windward slopes and ridgetops and the lowest observed speeds occurred on the leeward side of the butte and in sheltered side drainages on the butte itself. Downscaling with WindNinja offers improved predictions at these locations as indicated by Fig. 2 (regression lines in closer proximity to the line of agreement) and Fig. 3 (spatial variability in predictions more closely matches that of the observations).

Additionally, the downscaled NAM wind speeds were as accurate as the downscaled HRRR and WRF-UW wind speeds (Fig. 2). This indicates that the NAM forecast was able to capture the important large-scale flow features around BSB such that the additional resolution provided by HRRR and WRF-UW was not essential to resolve additional flow features in the large scale flow around BSB.

The accuracy of the NAM forecast at BSB is likely due to the fact that Snake River Plain which surrounds BSB is relatively flat and extends more than 50 km in all directions from the butte. Even a 12 km grid resolution would be capable of resolving the Snake River Plain and diurnal flow patterns within this large, gentle-relief drainage. Coarse-resolution models would not be expected to offer this same level of accuracy in areas of more extensive complex terrain, however. In areas surrounded by highly complex terrain it may be necessary to acquire NWP model output on finer grids in order to resolve the regional flow features.

The NWP forecasts predicted the overall temporal trend in wind speed (Fig. 3), but underestimated peak wind speeds due to under predictions on ridgetops and windward slopes as previously discussed, and also occasionally in the flat terrain on the Snake River Plain surrounding the butte (Fig. 4).

NWP models with coarser resolution grids predicted less spatial variability in wind speed (Fig. 3). This is because there were fewer grid cells covering the domain, and

**ACPD**

doi:10.5194/acp-2015-761

**Downscaling surface wind predictions with WindNinja**

N. S. Wagenbrenner et al.

thus fewer prediction points around the butte. The spatial variability in the downscaled wind speed predictions more closely matched that of the observed data, although the highest speeds were still under predicted (Fig. 3). Although downscaling generally improved the spatial variability of the predictions, there were cases where NWP errors clearly propagated into the downscaled simulations. For example, HRRR frequently over predicted morning wind speeds associated with down-drainage flow on the Snake River Plain; this error was amplified in the downscaled simulations, especially at the ridgetop locations (e.g., Figs. 3 and 4, 15–17 July).

The mean bias, RMSE, and SDE for wind speed and wind direction were smaller in nearly all cases for the downscaled simulations than for the NWP forecasts during the five-day period (Table 2). Mean biases in wind speed were all slightly negative and NAM and WRF-UW had the largest mean biases. The RMSE and SDE in wind speed were largest for HRRR. Although mean bias, RMSE, and SDE in wind direction for the downscaled forecasts were smaller or equal to those for the NWP forecasts, the differences were small, with a maximum reduction in mean bias in wind direction of just 4°.

It is difficult to draw too many conclusions from the spatially and temporally averaged 5 day statistics, however, since this period included a range of meteorological conditions (e.g., high-wind events from different directions, upslope flow, downslope flow) each of which could have been predicted with a different level of skill by the models. Qualitatively, however, the 5 day results demonstrate that the spatial variability in the downscaled winds better matches that of the observed winds at BSB (Fig. 3) and, although the reductions were small in some cases, nearly all statistical metrics also improved with downscaling. The analysis is broken down by flow regime in the next section for more insight into model performance.

## 4.2 Performance under Upslope, downslope, and externally-forced flows

Local solar heating and cooling was a primary driver of the flow during the slope flow regime at BSB (Butler et al., 2015), with local thermal effects equal to or exceeding

**ACPD**

doi:10.5194/acp-2015-761

**Downscaling surface wind predictions with WindNinja**

N. S. Wagenbrenner et al.

the local mechanical effects of the terrain on the flow. Because there is weak external forcing (i.e., input wind speeds to WindNinja are low), the downscaling is largely driven by the diurnal slope flow parameterization in WindNinja during the slope flow regimes.

During upslope flow, the diurnal slope flow parameterization increases speeds on the windward slopes and reduces speeds (or reverses flow and increases speeds, depending on the strength of the slope flow relative to the prevailing flow) on lee slopes due to the opposing effects of the prevailing wind and the thermal slope flow. The parameterization has the opposite effect during downslope flow; windward slope speeds are reduced (or possibly increased if downslope flow is strong enough to reverse the prevailing flow) and lee side speeds are enhanced.

### 4.2.1 Wind speed

The biggest improvements in wind speed predictions from downscaling occurred during externally-driven flow events (Fig. 5). This is not surprising since the highest spatial variability in the observed wind speeds occurred during high-wind events due to mechanically-induced effects of the terrain, with higher speeds on ridges and windward slopes and lower speeds in sheltered side drainages and on the lee side of the butte (Figs. 6–8). Since WindNinja is designed primarily to simulate the mechanical effects of the terrain on the flow, it is during these high-wind events that the downscaling has the most opportunity to improve predictions across the domain.

The NWP models tended to under predict wind speeds on the windward slopes, ridgetops, and surrounding flat terrain, and over predict on the lee side of the butte during high wind events (e.g., Fig. 6). The largest NWP errors in wind speed during high wind events were on the ridgetops, where speed-up occurred and the NWP under predicted speeds. These largest wind speed errors were reduced by downscaling (e.g., Fig. 6). Downscaling reduced NWP wind speed errors in most regions on the butte, although the general trend of under predicting wind speeds on the windward side and over predicting on the lee side did not change (e.g., Fig. 6).

Discussion Paper | Discussion Paper | Discussion Paper | Discussion Paper | Discussion Paper

**ACPD**

doi:10.5194/acp-2015-761

**Downscaling surface wind predictions with WindNinja**

N. S. Wagenbrenner et al.

There were consistent improvements in predicted wind speeds from downscaling during the upslope regime, although the improvements were smaller than for the externally-driven regime (Fig. 5). Wind speeds were lower during the slope flow regimes than during the externally-forced regime (Figs. 6–8), and thus, smaller improvements were possible with downscaling. There was some speed-up predicted on the windward side of the butte during the representative upslope case which appeared to match the observed wind field (Fig. 8).

Results were mixed for the downslope regime, as wind speeds improved with downscaling for WRF-UW and NAM, but not for WRF-NARR or HRRR (Fig. 5). The poor wind speed predictions from HRRR during the downslope regime is partly due to the fact that HRRR tended to over predict early morning winds associated with down drainage flows on the Snake River Plain. These errors were amplified by the downscaling, especially at ridgetop locations (Fig. 4). In reality, the high-elevation ridgetop locations tended to be decoupled from lower-level surface winds during the slope flow regimes due to flow stratification. WindNinja assumes neutral atmospheric stability, however, so this stratification is not handled. A parameterization for non-neutral atmospheric conditions is currently being tested in Windninja.

The diurnal slope flow parameterization in WindNinja resulted in lower speeds on the windward side and higher speeds on the lee side of the butte for the representative downslope case (Fig. 7). These downscaled speeds better matched those of the observed wind field, although speeds were still under predicted for ridgetops and a few other locations around the butte (Fig. 7). The high observed speeds at the ridgetop locations are not likely due to thermal slope flow effects, but could be from the influence of gradient-level winds above the nocturnal boundary layer. These ridgetop locations are high enough in elevation (800 m above the surrounding plain) that they likely protruded out of the nocturnal boundary layer and were exposed to the decoupled gradient-level winds. Butler et al. (2015) noted that ridgetop winds did not exhibit a diurnal pattern and tended to be decoupled from winds at other locations on and around the butte. Lack of diurnal winds at the summit of the butte is also confirmed

**ACPD**

doi:10.5194/acp-2015-761

**Downscaling surface wind predictions with WindNinja**

N. S. Wagenbrenner et al.

by National Oceanic and Atmospheric Administration Field Research Division (NOAA-FRD) mesonet station data collected at the top of BSB (described in Butler et al., 2015; http://www.noaa.inel.gov/projects/INLMet/INLMet.htm).

Under predictions on the lower slopes and on the plain surrounding the butte could be due to overly weak slope flows being generated by the slope flow parameterization in WindNinja (Figs. 7 and 8). Overly weak slope flows could be caused by a number of things: improper parameterization of surface or entrainment drag parameters, poor estimation of the depth of the slope flow, or deficiencies in the micrometeorological model used. The slope flow parameterization is being evaluated in a companion paper.

## 4.2.2   Wind direction

The biggest improvement in wind direction predictions from downscaling occurred during the downslope regime (Fig. 5). Wind direction improved with downscaling for all NWP models during periods of downslope flow. This indicates that the diurnal slope flow model helped to orient winds downslope. This is confirmed by inspection of the vector plots for the representative downslope case which show the downscaled winds oriented downslope on the southwest and northeast faces of the butte (Fig. 7). Downscaling reduced speeds on the northwest (windward) side of the butte, but did not predict strong enough downslope flow in this region to reverse the flow from the prevailing northwest direction (Fig. 7). This again suggests that perhaps the diurnal slope flow algorithm is predicting overly weak slope flows.

Wind direction predictions during the upslope regime also improved with downscaling for all NWP models except HRRR (Fig. 5). Downscaled winds for the representative upslope case were oriented upslope on the southwest (lee side) of the butte and matched the observed winds in this region well (Fig. 8). This is an improvement over the NWP wind directions on the lee side of the butte.

There was no improvement in wind direction predictions with downscaling during the externally-driven regime (Fig. 5). Looking at the vector plots during the representative externally-driven event (Fig. 6), it is clear why this would be. The representative event

**ACPD**

doi:10.5194/acp-2015-761

**Downscaling surface wind predictions with WindNinja**

N. S. Wagenbrenner et al.

was a high-wind event from the southwest. Wind directions are well predicted on the windward side of the butte, but not on the leeward side, where the observed field indicates some recirculation in the flow field (Fig. 6). The prevailing southwesterly flow is captured by the NWP model, but the lee side recirculation is not. WindNinja does not predict the lee side recirculation, and thus, the downscaling does not improve directions on the lee side of the butte (Fig. 7). This is an expected result, as WindNinja has been shown to have difficulties simulating flows on the lee side of terrain features due to the fact that it does not account for conservation of momentum in the flow solution (Forthofer et al., 2014a).

## 5  Summary

Results showed that the NWP models captured the important large-scale flow features around BSB under most conditions, but were not capable of predicting the high spatial variability (scale of 100s of meters) in the observed winds on and around the butte induced by mechanical effects of the terrain and local surface heating and cooling. Thus, surface winds from the NWP models investigated in this study would not be sufficient for forecasting wind speeds on and around the butte at the spatial scales relevant for processes driven by local surface winds, such as wildland fire spread, for example.

Wind predictions generally improved for all NWP models by downscaling with Wind-Ninja. The biggest improvements occurred under high-wind events (near-neutral atmospheric stability) when observed wind speeds were greater than $10 \, \mathrm{m \, s^{-1}}$. Downscaled NAM wind speeds were as accurate as downscaled WRF-UW and HRRR wind speeds, indicating that a NWP model with 12 km grid resolution was sufficient for capturing the large-scale flow features around BSB.

WindNinja did not predict the observed lee-side flow recirculation at BSB that occurred during externally-forced high wind events. Previous work has shown that Wind-Ninja has difficulties simulating lee-side flows (Forthofer et al., 2014a). This is partly

**ACPD**

doi:10.5194/acp-2015-761

**Downscaling surface wind predictions with WindNinja**

N. S. Wagenbrenner et al.

due to lack of a momentum equation in the WindNinja flow solution as discussed in Forthofer et al. (2014a). Work is currently underway to incorporate an optional momentum solver in WindNinja which is anticipated to improve flow predictions on the lee-side of terrain obstacles.

Results indicated that WindNinja predicted overly weak slope flows compared to observations. Weak slope flow could be caused by several different issues within the diurnal slope flow parameterization in WindNinja: improper parameterization of surface or entrainment drag parameters, poor estimation of the depth of the slope flow, or deficiencies in the micrometeorological model. These issues will be explored in future work.

This work constitutes evaluation of a diagnostic wind model at unprecedented high spatial resolution and terrain complexity. While extensive evaluations have been performed with data collected in less rugged terrain (e.g., Askervein Hill and Bolund Hill, relatively low elevation hills with simple geometry), to our knowledge, this study is the first to evaluate a diagnostic wind model with data collected in terrain with topographical ruggedness approaching that of typical landscapes in the western US susceptible to wildland fire. This work demonstrates that NWP model wind forecasts can be improved in complex terrain, at least in some cases, through dynamic downscaling via a mass-conserving wind model. These improvements should propagate on to more realistic predictions from other model applications which are sensitive to surface wind fields, such as wildland fire behavior, local-scale transport and dispersion, and wind energy applications.

*Acknowledgements.* Thanks to Dave Ovens and Cliff Mass of the University of Washington for providing access to the WRF-UW simulations and Eric James of NOAA–GSD Earth System Research Laboratory for access to the HRRR simulations. Thanks to Serena Chung of the Laboratory for Atmospheric Research, Washington State University, for guidance on the WRF-NARR simulations. We also thank the participants in the BSB field campaigns, including Dennis Finn, Dan Jimenez, Paul Sopko, Mark Vosburgh, Larry Bradshaw, Cyle Wold, Jack Kautz, and Randy Pryhorocki.

**ACPD**

doi:10.5194/acp-2015-761

**Downscaling surface wind predictions with WindNinja**

N. S. Wagenbrenner et al.

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

**Table 1.** Model specifications.

| Model | Horizontal grid resolution | Number vertical layers | First layer height* (m a.g.l.) | Top height* (m a.g.l.) | Numerical core | Run frequency |
| --- | --- | --- | --- | --- | --- | --- |
| NAM | 12 km | 26 | 200 | 15 000 | NMM | 00z, 06z, 12z, 18z |
| WRF-UW | 4 km | 38 | 40 | 16 000 | ARW | 00z, 12z |
| HRRR | 3 km | 51 | 8 | 16 000 | ARW | hourly |
| WRF-NARR | 1.33 km | 33 | 38 | 15 000 | ARW | NA |
| WindNinja | 138 m | 20 | 1.92 | 931 | NA | NA |

* Approximate average height a.g.l.

**Table 2.** Model mean bias, root-mean-square error (RMSE), and standard deviation of errors (SDE) for surface wind speeds and directions during the 5 day evaluation period at Big Southern Butte. Downscaled values are in parentheses. Smaller values are in bold. The 5 day period includes the Downslope, Upslope, and Externally-driven time periods.

| Time period | Statistic | NAM | WRF-UW | HRRR | WRF-NARR |
|---|---|---|---|---|---|
| | | Wind Speed (m s$^{-1}$) | | | |
| 5-day | Bias | −0.84 **(−0.67)** | −1.17 **(−0.95)** | −0.40 **(−0.14)** | −0.31 **(−0.08)** |
| | RMSE | 2.31 **(2.04)** | 2.39 **(2.07)** | 2.52 **(2.47)** | 2.33 **(2.21)** |
| | SDE | 2.15 **(1.92)** | 2.08 **(1.83)** | 2.49 **(2.47)** | 2.31 **(2.21)** |
| Downslope | Bias | −1.07 **(−0.76)** | −1.15 **(−0.74)** | **−0.09** (0.48) | −0.48 **(0.12)** |
| | RMSE | 2.08 **(1.92)** | 2.03 **(1.83)** | **2.36** (2.66) | **2.19** (2.28) |
| | SDE | 1.79 **(1.77)** | **1.67** (1.68) | **2.36** (2.62) | **2.14** (2.28) |
| Upslope | Bias | −0.81 **(−0.74)** | −1.11 **(−0.98)** | −0.81 **(−0.75)** | 0.06 **(0.05)** |
| | RMSE | 1.73 **(1.62)** | 2.02 **(1.86)** | 1.93 **(1.81)** | 1.86 (1.86) |
| | SDE | 1.52 **(1.44)** | 1.69 **(1.58)** | 1.76 **(1.64)** | 1.86 (1.86) |
| Externally-driven | Bias | **−0.57** (−0.62) | **−1.28** (−1.32) | **−0.94** (−1.03) | **−0.22** (−0.33) |
| | RMSE | 3.06 **(2.48)** | 3.21 **(2.58)** | 3.17 **(2.59)** | 2.92 **(2.39)** |
| | SDE | 3.00 **(2.40)** | 2.94 **(2.22)** | 3.02 **(2.38)** | 2.92 **(2.37)** |
| | | Wind Direction (°) | | | |
| 5-day | Bias | 59 **(56)** | 57 **(53)** | 64 **(60)** | 57 **(54)** |
| | RMSE | 76 **(72)** | 74 **(71)** | 80 **(76)** | 73 **(71)** |
| | SDE | 47 **(46)** | 47 **(46)** | 47 **(46)** | 46 (46) |
| Downslope | Bias | 67 **(60)** | 61 **(56)** | 76 **(67)** | 66 **(61)** |
| | RMSE | 83 **(77)** | 78 **(72)** | 88 **(81)** | 81 **(75)** |
| | SDE | 49 **(47)** | 48 **(46)** | 46 (46) | 47 **(45)** |
| Upslope | Bias | 55 **(52)** | 58 **(54)** | 56 (56) | 52 **(49)** |
| | RMSE | 70 **(67)** | 74 **(71)** | 72 (72) | 68 **(65)** |
| | SDE | 44 **(42)** | 46 **(45)** | 45 (46) | 44 **(42)** |
| Externally-driven | Bias | **48** (49) | **45** (46) | 51 **(50)** | **44** (46) |
| | RMSE | **64** (65) | **63** (65) | 68 **(67)** | **62** (65) |
| | SDE | **43** (44) | **44** (47) | 45 **(44)** | **43** (46) |

Discussion Paper | Discussion Paper | Discussion Paper | Discussion Paper

**ACPD**

doi:10.5194/acp-2015-761

**Downscaling surface wind predictions with WindNinja**

N. S. Wagenbrenner et al.

Discussion Paper | Discussion Paper | Discussion Paper | Discussion Paper |

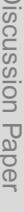

# ACPD

doi:10.5194/acp-2015-761

**Downscaling surface wind predictions with WindNinja**

N. S. Wagenbrenner et al.

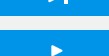



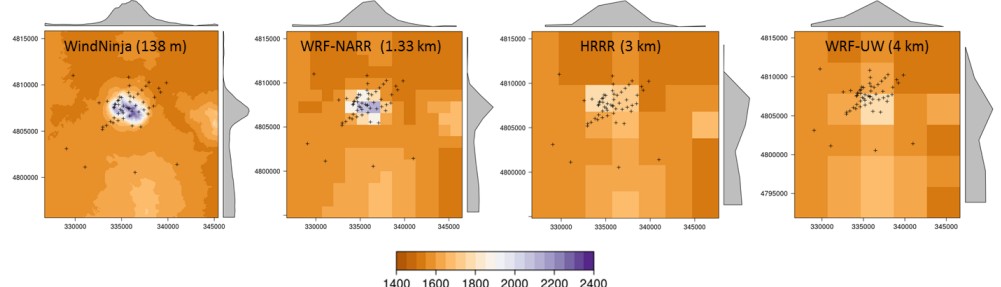

**Figure 1.** Terrain representation (m a.s.l.) in WindNinja, WRF-NARR, HRRR, and WRF-UW for the Big Southern Butte. Crosses indicate surface sensor locations. Maps are projected in UTM zone 12. Axis labels are eastings and northings in m. Profiles in gray are the average elevations for rows and columns in the panel. NAM (12 km) terrain is represented by just four cells and is not shown here.

Discussion Paper | Discussion Paper | Discussion Paper | Discussion Paper |

**ACPD**

doi:10.5194/acp-2015-761

**Downscaling surface wind predictions with WindNinja**

N. S. Wagenbrenner et al.

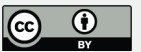

**Figure 2.** Observed vs. predicted wind speeds for the 5 day evaluation period at Big Southern Butte. Dashed black line is the 1 : 1 line. Colored lines are linear regressions (quadratic fit); dashed lines are NWP models and solid lines are NWP forecasts downscaled with WindNinja. Shading indicates 95 % confidence intervals.

# ACPD

doi:10.5194/acp-2015-761

**Downscaling surface wind predictions with WindNinja**

N. S. Wagenbrenner et al.

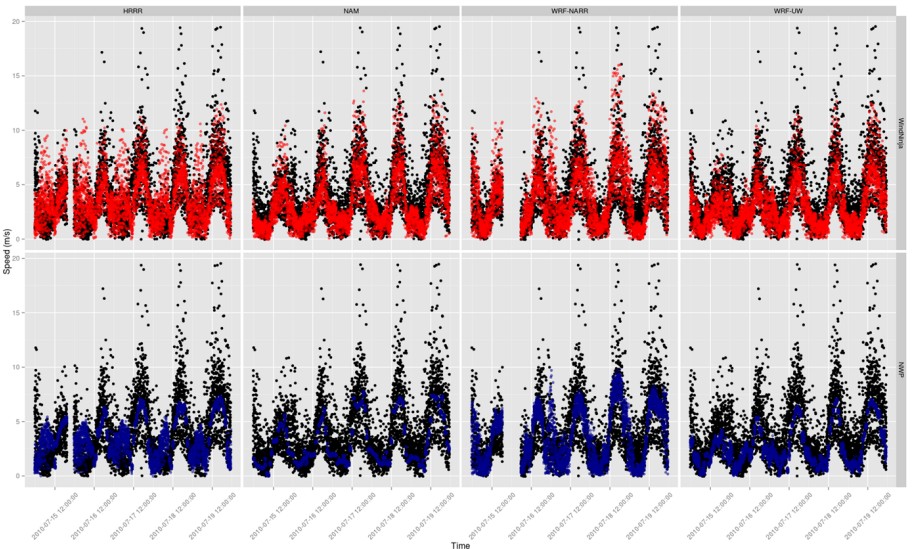

**Figure 3.** Observed (black) and predicted (colored) winds speeds at all sensors for 15–19 July 2010 at Big Southern Butte. Top panels are WindNinja predictions. Bottom panels are NWP predictions.

Discussion Paper | Discussion Paper | Discussion Paper | Discussion Paper | Discussion Paper

**ACPD**

doi:10.5194/acp-2015-761

**Downscaling surface wind predictions with WindNinja**

N. S. Wagenbrenner et al.

**Figure 4.** Observed (black line) and predicted (colored lines) wind speeds for sensor R2 located 5 km southwest of Big Southern Butte on the Snake River Plain and sensor R26 located on a ridgetop. Dashed colored lines are NWP and solid colored lines are WindNinja.

ACPD

doi:10.5194/acp-2015-761

**Downscaling surface wind predictions with WindNinja**

N. S. Wagenbrenner et al.

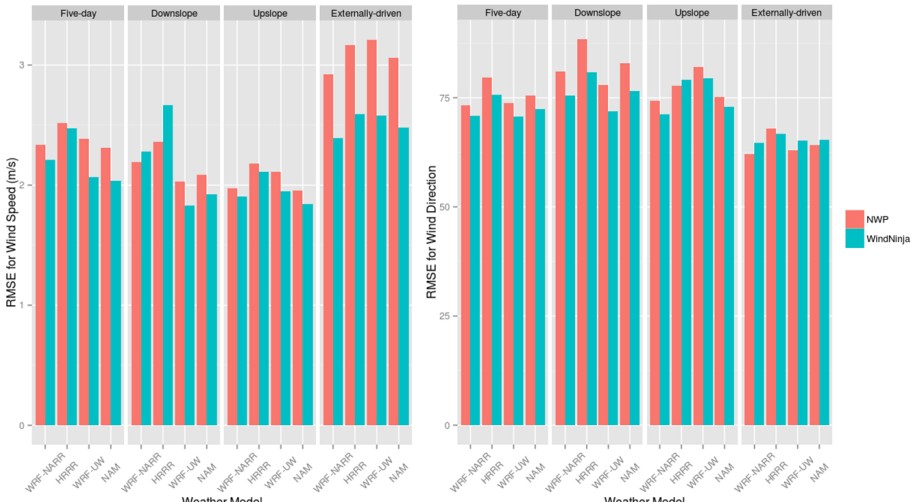

**Figure 5.** Root-mean-square error in wind speed (left) and wind direction (right) at Big Southern Butte for the five-day evaluation period ($N = 4149$), and downslope ($N = 1593$), upslope ($N = 717$), and externally-driven ($N = 966$) periods within the five-day period. Sample size, $N$ = number of hours × number of sensor locations.

Title Page

Abstract    Introduction

Conclusions    References

Tables    Figures

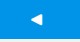    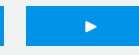

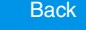    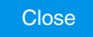



Discussion Paper | Discussion Paper | Discussion Paper | Discussion Paper

# ACPD

doi:10.5194/acp-2015-761

**Downscaling surface wind predictions with WindNinja**

N. S. Wagenbrenner et al.

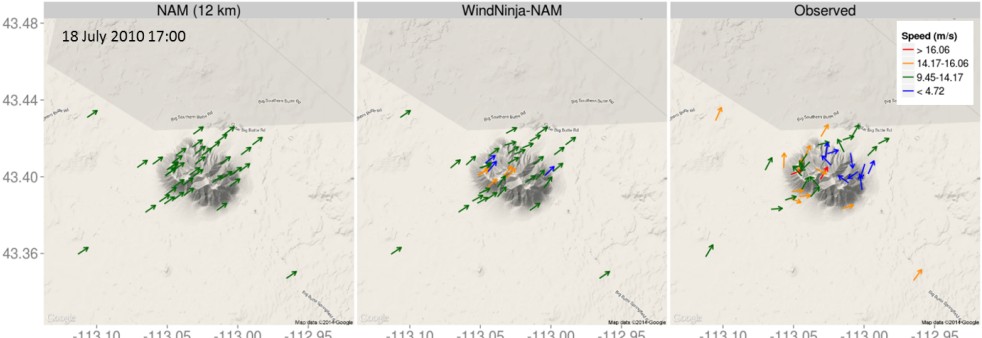

**Figure 6.** Predicted and observed winds for an externally-forced flow event at Big Southern Butte.

Discussion Paper | Discussion Paper | Discussion Paper | Discussion Paper | Discussion Paper

# ACPD

doi:10.5194/acp-2015-761

**Downscaling surface wind predictions with WindNinja**

N. S. Wagenbrenner et al.

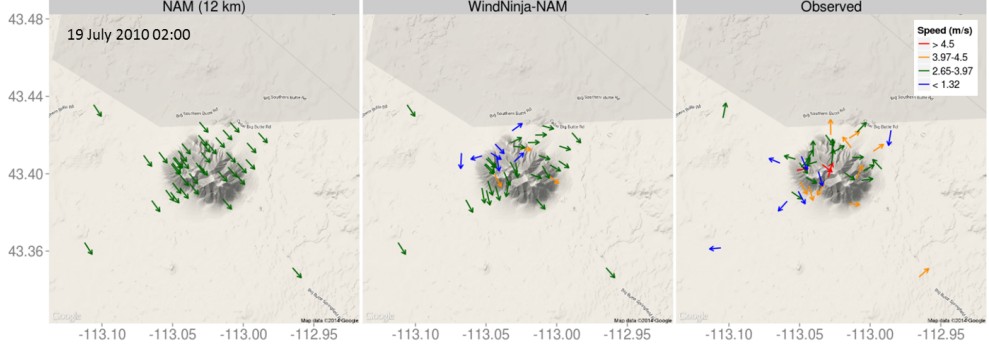

**Figure 7.** Predicted and observed winds for a downslope flow event at Big Southern Butte.

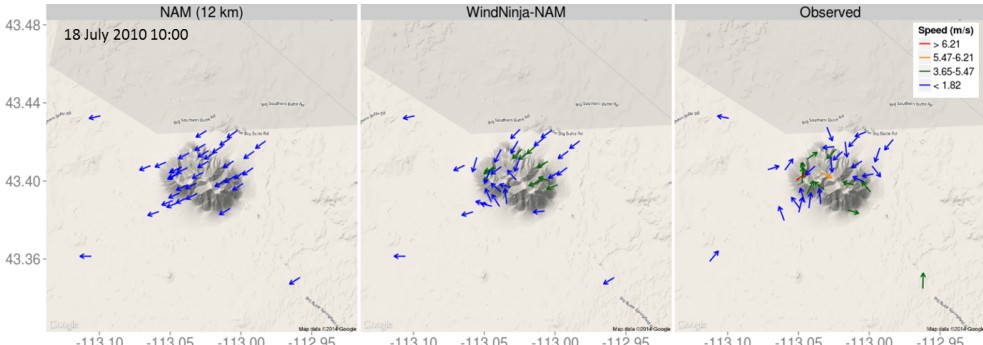

**Figure 8.** Predicted and observed winds for an upslope flow event at Big Southern Butte.

# ACPD

doi:10.5194/acp-2015-761

**Downscaling surface wind predictions with WindNinja**

N. S. Wagenbrenner et al.