# Peer review of "Downscaling Surface Wind Predictions from Numerical Weather Prediction Models in"

_Atmospheric Chemistry and Physics, 2015_

## Referee Comment (RC1) · Anonymous Referee #1 · 21 Feb 2016

Review

This is very well-written and well-constructed, scientifically sound paper that is of benefit to the scientific community. I don't have any major issues with the paper as-is. However, some discussion of the following issues would strengthen the paper in my opinion:

1. For Figures 1, and 6-8 it would be really helpful to have the Big Southern Butte area zoomed in more. I have a hard time seeing what is going on in that region and then the rest of the plots are mostly empty with a few observations scattered downstream. What I would recommend would be to have a zoomed-in plot of the Big Southern Butte area.

[Figure]

2. On the same note, it would be helpful if sensor R2, R26 discussed in Fig. 4 were indicated on the map in Fig. 1.

3. Why is the only meteorological parameter discussed in this paper wind? Is there no parallel for downscaling for other critical fire parameter: relative humidity and temperature? For the general reader some discussion of this in the intro would be helpful for the more general audience of ACP.

4. I am under the impression that strongly-forced wind events are the most important factor for high fire spread, and therefore the strengths of this study play to that need. I think this point should be made somewhere, even if it is fairly obvious.

5. WRF-HRRR fields are now available to drive the 1.33 km model. Were any tests conducted to determine the improvement in WRF using HRRR analyses to drive the 1.33 km model instead of NARR? Could this be included as potential future work?

6. I would separate the discussion more clearly into 'externally forced flows' (large-scale winds) and locally-forced flows (upslope and downslope). The current text discusses weak external forcing in many places; my opinion is that it would be better to distinguish the two cases by referring to 'locally-forced' more when the external forcing is weak.

7. Why was a 'quasi-large-eddy simulation' (200-400 m horizontal grid spacing) not conducted with WRF and compared to the other runs? This simply requires turning off the PBL scheme and a few other parameters (WRF-LES settings). A lack of numerical resources is a good argument. While the argument is made that these simulations are not feasible in real-time, the 1.33 km runs are also not feasible in real-time. If the LES run was conducted, then a comparison could be made such as 'the downscaling is equivalently good as the quasi-LES run over the area where the terrain structure is explicitly simulated. There may be issues with CFL criteria with high resolution topography in the simulations, but many investigators are starting to use WRF as a LES model with the PBL turned off…would be curious how these results would compare

to the downscaled. In any case, a sentence or two in future work discussing how this could be done in the future would be good.

8. The potential limitations of the slope flow parameterization and discussion of a companion paper is looking at this is discussed but in a rather scattered manner. If a summary of the strengths and weaknesses of WindNinja were included in the introduction this would help make the findings in the body of the paper flow better.

9. The weakness with windninja simulating lee-side flows is an important point, as many fires occur on the 'downslope' side of mountain ranges. Does this weakness also happen during strongly forced conditions? It was not clear to me.

---

## Referee Comment (RC2) · Anonymous Referee #2 · 23 Feb 2016

The authors compare different WRF simulations and a diagnostic model against wind observations over an isolated mountain. Spacial emphasis is given to quantify the added value of downscaling the WRF simulations with the diagnostic model (Wind-Ninja). The topic is relevant to progress in our understanding of the ability of mesoscale simulations, and diagnostic models, to reproduce the wind over complex terrain. The observational dataset is quite unique. The article is well written and the conclusions are supported by the results. Perhaps it could be highlighted that none of the WRF simulations is able to represent the mountain. Only the simulation at 1.33 km of horizontal resolution starts to "see" something (fig. 1). More specific comments, all of them of minor character, are provided below.

[Figure]

I therefore recommend acceptance of the manuscript pending to minor revisions

SPECIFIC COMMENTS

1. Page 5, line 9: Please define the FDM acronym.

2. Page 5, line 25: I believe you are using the Kain-Fritsch scheme to represent the cumulus in WRF. If so, please update the reference.

3. Page 6, line 4: The first model level of WRF-UWis 40m a.g.l. Later in the manuscript it is mentioned that these winds are interpolated to 3 m using a log profile. Is this interpolation taking into account the atmospheric stability? Or the atmosphere is assumed to be neutral? I say this because the target height, 3 m a.g.l, is far from 40 m a.g.l. and the interpolation method could play a major role in the evaluation.

4. Page 7, line 7: The NAM model has the first level at 200 m and the horizontal resolution is 12 km. NAM is obviously not representing the mountain at all. RAP provides information of the winds in the valley. It would be informative to have some discussion of this here pointing out why NAM is used even though the mountain is not represented at all.

5. Page 13: liens 11-15: None of the WRF simulations have enough horizontal resolution to represent the mountain. Only at 1.3 km WRF starts to see something. The WRF predictability probably comes from the simulation of the flow in the Snake River Plain. It may be interesting to compute anomalies with respect to the mean flow and compare them with the equivalent observations to see if there is any benefit as a result of increasing the horizontal resolution.

6. Page 14: It will be good to show a spatial plot showing the under prediction of windward and ridge top locations.

7. Page 18, first paragraph Summary: You should mention that the horizontal resolutions are too coarse to represent the mountain in the WRF simulations.

---

## Author Comment (AC1) · 14 Mar 2016

Reply to Referee #1

Thanks for the comments and suggestions. These will strengthen the paper. Here are our detailed responses:

1. Figures 1 and 6-8 will be updated to show a zoomed in version of the butte.

2. R2 and R26 will be added to Figure 1.

3. The diagnostic model evaluated in this paper, WindNinja, is only designed to downscale the flow. WindNinja includes physics for modeling the mechanical and thermal

effects of the terrain on the flow field. WindNinja is capable of interpolating other parameters (e.g., temperature and relative humidity) to a finer grid, but does not provide any additional physics (e.g., conservation of energy) or parameterizations to simulate terrain effects on these parameters. For these reasons, WindNinja does not output additional downscaled weather parameters. Additionally, wind varies more spatially than temperature and RH, so is more important to predict at a high resolution. Wind is known to often be the driving environmental variable for wildfire spread and behavior. We will clarify these points in the paper.

4. Yes, it is correct that high winds are often the most important factor for wildfire spread. This point will be incorporated into the paper.

5. HRRR-initialized 1.33 km WRF runs were not considered in this study, but could be considered in the future.

6. The discussion will be adjusted accordingly to more clearly separate the externally-forced flow and locally-forced flow discussion.

7. LES was not considered for a couple of reasons. Most importantly, LES is too computationally intensive to be used in an operational context in an emergency response situtation such as wildland fire. Additionally, there appear to still be many issues regarding LES in complex terrain. For example, as we understand it, WRF-LES cannot be run in complex terrain with the typical meshing algorithm employed by WRF; instead some other method, such as IBM must be used. Becasue of these issues, LES was not considered. However, we are working with colleagues who have substantial experience with LES that are investigating LES simulations at Big Southern Butte. We plan to make comparisons between WindNinja, the next generation WindNinja with a RANS-based solver added, and these LES simulations in the future.

8. The discussion of the slope flow parameterization will be re-worked. We will also include some background information in the introduction to set the stage for this discussion.

9. Yes, the weakness in simulating lee-side recirculation occurs under high wind speeds as well. We will re-work this discussion to clarify the lee-side flow behavior and difficulty in simulating that behavior.

---

## Author Comment (AC2) · 14 Mar 2016

Reply to Referee #2

Thanks for the comments and suggestions. We will highlight the lack of terrain representation in the WRF forecasts as suggested. Here are our detailed responses:

1. FDM will be defined.

2. Yes, this reference will be added.

3. The interpolation assumes neutral atmospheric stability. This information will be added in the methods.

4. We will include additional discussion of the terrain representation in NAM and its inability to resolve the butte.

5. Yes, looking at the perturbations to the mean flow could be an interesting addition to our analysis. We will consider adding this in the revised manuscript.

6. We will consider adding a spatial plot of the bias at the windward and ridgetop locations.

7. We will include discussion of the horizontal resolution and terrain representation in the summary.

---

## Author Response (AR2)

Review

This is very well-written and well-constructed, scientifically sound paper that is of benefit to the scientific community. I don't have any major issues with the paper as-is. However, some discussion of the following issues would strengthen the paper in my opinion:

1. For Figures 1, and 6-8 it would be really helpful to have the Big Southern Butte area zoomed in more. I have a hard time seeing what is going on in that region and then the rest of the plots are mostly empty with a few observations scattered downstream. What I would recommend would be to have a zoomed-in plot of the Big Southern Butte area.

[Figure]

2. On the same note, it would be helpful if sensor R2, R26 discussed in Fig. 4 were indicated on the map in Fig. 1.

3. Why is the only meteorological parameter discussed in this paper wind? Is there no parallel for downscaling for other critical fire parameter: relative humidity and temperature? For the general reader some discussion of this in the intro would be helpful for the more general audience of ACP.

4. I am under the impression that strongly-forced wind events are the most important factor for high fire spread, and therefore the strengths of this study play to that need. I think this point should be made somewhere, even if it is fairly obvious.

5. WRF-HRRR fields are now available to drive the 1.33 km model. Were any tests conducted to determine the improvement in WRF using HRRR analyses to drive the 1.33 km model instead of NARR? Could this be included as potential future work?

6. I would separate the discussion more clearly into 'externally forced flows' (large-scale winds) and locally-forced flows (upslope and downslope). The current text discusses weak external forcing in many places; my opinion is that it would be better to distinguish the two cases by referring to 'locally-forced' more when the external forcing is weak.

7. Why was a 'quasi-large-eddy simulation' (200-400 m horizontal grid spacing) not conducted with WRF and compared to the other runs? This simply requires turning off the PBL scheme and a few other parameters (WRF-LES settings). A lack of numerical resources is a good argument. While the argument is made that these simulations are not feasible in real-time, the 1.33 km runs are also not feasible in real-time. If the LES run was conducted, then a comparison could be made such as 'the downscaling is equivalently good as the quasi-LES run over the area where the terrain structure is explicitly simulated. There may be issues with CFL criteria with high resolution topography in the simulations, but many investigators are starting to use WRF as a LES model with the PBL turned off...would be curious how these results would compare

to the downscaled. In any case, a sentence or two in future work discussing how this could be done in the future would be good.

8. The potential limitations of the slope flow parameterization and discussion of a companion paper is looking at this is discussed but in a rather scattered manner. If a summary of the strengths and weaknesses of WindNinja were included in the introduction this would help make the findings in the body of the paper flow better.

9. The weakness with windninja simulating lee-side flows is an important point, as many fires occur on the 'downslope' side of mountain ranges. Does this weakness also happen during strongly forced conditions? It was not clear to me.
* * *
[Figure]

Atmos. Chem. Phys. Discuss.,
doi:10.5194/acp-2015-761-RC2, 2016

[Figure]
The authors compare different WRF simulations and a diagnostic model against wind observations over an isolated mountain. Spacial emphasis is given to quantify the added value of downscaling the WRF simulations with the diagnostic model (Wind-Ninja). The topic is relevant to progress in our understanding of the ability of mesoscale simulations, and diagnostic models, to reproduce the wind over complex terrain. The observational dataset is quite unique. The article is well written and the conclusions are supported by the results. Perhaps it could be highlighted that none of the WRF simulations is able to represent the mountain. Only the simulation at 1.33 km of horizontal resolution starts to "see" something (fig. 1). More specific comments, all of them of minor character, are provided below.

[Figure]

I therefore recommend acceptance of the manuscript pending to minor revisions

SPECIFIC COMMENTS

1. Page 5, line 9: Please define the FDM acronym.

2. Page 5, line 25: I believe you are using the Kain-Fritsch scheme to represent the cumulus in WRF. If so, please update the reference.

3. Page 6, line 4: The first model level of WRF-UWis 40m a.g.l. Later in the manuscript it is mentioned that these winds are interpolated to 3 m using a log profile. Is this interpolation taking into account the atmospheric stability? Or the atmosphere is assumed to be neutral? I say this because the target height, 3 m a.g.l, is far from 40 m a.g.l. and the interpolation method could play a major role in the evaluation.

4. Page 7, line 7: The NAM model has the first level at 200 m and the horizontal resolution is 12 km. NAM is obviously not representing the mountain at all. RAP provides information of the winds in the valley. It would be informative to have some discussion of this here pointing out why NAM is used even though the mountain is not represented at all.

5. Page 13: liens 11-15: None of the WRF simulations have enough horizontal resolution to represent the mountain. Only at 1.3 km WRF starts to see something. The WRF predictability probably comes from the simulation of the flow in the Snake River Plain. It may be interesting to compute anomalies with respect to the mean flow and compare them with the equivalent observations to see if there is any benefit as a result of increasing the horizontal resolution.

6. Page 14: It will be good to show a spatial plot showing the under prediction of windward and ridge top locations.

7. Page 18, first paragraph Summary: You should mention that the horizontal resolutions are too coarse to represent the mountain in the WRF simulations.

[Figure]

[Figure]

Reply to Reviewer 1

1. Figures 1 and 6-8 will be updated to show a zoomed in version of the butte.  Zooming in on the butte
would be nice for the left panel of Figure 1, however, it's not really possible for the other three panels,
since the butte is represented by just one or two pixels; therefore, we chose to leave these figures as is.
Additionally, these figures depict the domain extent used in our downscaling simulations and so there is
value in leaving extents in the figures as is.

2. R2 and R26 will be added to Figure 1.  Added in Figure 1, p. 34.

3. The diagnostic model evaluated in this paper, WindNinja, is only designed to downscale the flow.
WindNinja includes physics for modeling the mechanical and thermal effects of the terrain on the flow
field. WindNinja is capable of interpolating other parameters (e.g., temperature and relative humidity)
to a finer grid, but does not provide any additional physics (e.g., conservation of energy) or
parameterizations to simulate terrain effects on these parameters. For these reasons, WindNinja does
not output additional downscaled weather parameters. Additionally, wind varies more spatially than
temperature and RH, so is more important to predict at a high resolution. Wind is known to often be the
driving environmental variable for wildfire spread and behavior. We will clarify these points in the
paper.  Some discussion on this was added in lines 101-104.

4. Yes, it is correct that high winds are often the most important factor for wildfire spread. This point will
be incorporated into the paper. Added in lines 93-94, 384-385, 473-475.

5. HRRR-initialized 1.33 km WRF runs were not considered in this study, but could be considered in the
future.

6. The discussion will be adjusted accordingly to more clearly separate the externally-forced flow and
locally-forced flow discussion.  After reviewing this section, we decided to leave the organization as is.
We currently have sections formally separated into wind speed vs. wind direction and all data vs.
diurnal/externally-forced flows.  The discussion is organized by paragraph (no mixed discussion of
externally-forced/externally-weak flows in a paragraph), but we didn't feel it was necessary to add
another formal section heading to separate these.

7. LES was not considered for a couple of reasons. Most importantly, LES is too computationally
intensive to be used in an operational context in an emergency response situation such as wildland fire.
Additionally, there appear to still be many issues regarding LES in complex terrain. For example, as we
understand it, WRF-LES cannot be run in complex terrain with the typical meshing algorithm employed
by WRF; instead some other method, such as IBM must be used. Because of these issues, LES was not
considered. However, we are working with colleagues who have substantial experience with LES that are
investigating LES simulations at Big Southern Butte. We plan to make comparisons between WindNinja,
the next generation WindNinja with a RANS-based solver added, and these LES simulations in the future.

8. The discussion of the slope flow parameterization will be re-worked. We will also include some
background information in the introduction to set the stage for this discussion. More discussion was
added in the introduction in lines 93-98.

9. Yes, the weakness in simulating lee-side recirculation occurs under high wind speeds as well. We will
re-work this discussion to clarify the lee-side flow behavior and difficulty in simulating that behavior.

Reply to Reviewer 2

1. FDM will be defined. Added in line 117.

2. Yes, this reference will be added. Added in line 131.

3. The interpolation assumes neutral atmospheric stability. This information will be added in the methods. Added in lines 202-203.

4. We will include additional discussion of the terrain representation NAM and its inability to resolve the butte. Added in lines 163-166.

5. Yes, looking at the perturbations to the mean flow could be an interesting addition to our analysis.
We will consider adding this in the revised manuscript. We decided not to add this at this time, but will
consider this method in future evaluation work we have planned.

6. We will consider adding a spatial plot of the bias at the windward and ridgetop locations. We decided
not to add this, but will consider this type of plot in our future evaluation work.

7. We will include discussion of the horizontal resolution and terrain representation in the summary.
Added in lines 462-463.

[revised manuscript text omitted]